# ONLINE CONTINUAL LEARNING UNDER CONDITIONAL DOMAIN SHIFT

## ABSTRACT

Existing continual learning benchmarks often assume each task's training and test data are from the same distribution, which may not hold in practice. Towards making continual learning practical, in this paper, we introduce a novel setting of online continual learning under conditional domain shift, in which domain shift exists between training and test data of all tasks: $P^{tr}(X,Y) \neq P^{te}(X,Y)$, and the model is required to generalize to unseen domains at test time. To address this problem, we propose *Conditional Invariant Experience Replay (CIER)* that can simultaneously retain old knowledge, acquire new information, and generalize to unseen domains. CIER employs an adversarial training to correct the shift in $P(X,Y)$ by matching $P(X|Y)$, which results in an invariant representation that can generalize to unseen domains during inference. Our extensive experiments show that CIER can bridge the domain gap in continual learning and significantly outperforms state-of-the-art methods. We will release our benchmarks and implementation upon acceptance.

## 1 INTRODUCTION

Continual learning is a promising framework towards human-level intelligence by developing models that can continuously learn over time (Ring, 1997; Parisi et al., 2019). Unlike traditional learning paradigms, continual learning methods observe a continuum of tasks and have to simultaneously perform well on all tasks with limited access to previous data. Therefore, they have to achieve a good trade-off between retaining old knowledge (French, 1999) and acquiring new skills, which is referred to as the *stability-plasticity dilemma* (Abraham & Robins, 2005). Continual learning is not only a challenging research problem but also has tremendous impacts on many applications (Diethe et al., 2019). In particular, the deployed model may encounter new problems over time, and re-training every time new data arrive is infeasible, especially for large, complex neural networks.

Despite recent success, existing continual learning benchmarks assume the data of each class are drawn from the same distribution during training and testing, i.e. $P^{tr}(X,Y) = P^{te}(X,Y)$. This restrictive assumption is unlikely to hold in practice and prohibits the use of continual learning strategies in numerous real-world applications. For example, consider a continual learning agent already trained to recognize certain objects in the *indoor* environment; then, it moves to a new *outdoor* environment to acquire new skills. The agent may fail to recognize the learned objects because they are placed in a completely different background. Such an environment change is referred to as *domain shift* (Khosla et al., 2012; Long et al., 2015; Diethe et al., 2019), and is very natural for continual learning in practice (Diethe et al., 2019). To formalize the continual learning under domain shift problem, we need to consider the interaction between domain and the observation. In many computer vision applications, the causal structure is usually $Y \rightarrow X$, i.e., the object class is the cause for image features (Lopez-Paz et al., 2017). This setting ignores the effect of *conditional domain shift* (Zhang et al., 2013), where both the domain and the object class are the causes for the image features. That is, images may come from different domains, but they have the same label. Figure 1 shows the causal graph for continual learning under conditional domain shift problem and an illustrative example of the domain shift between training and testing data.

Continual learning under conditional domain shift requires the model to perform previous tasks in new domains, which poses a great challenge since we do not know in hindsight which domain will be tested. Therefore, the model needs to learn the concepts presented in the data while ignoring the domains. For example, placing an object in different background does not change the label of that

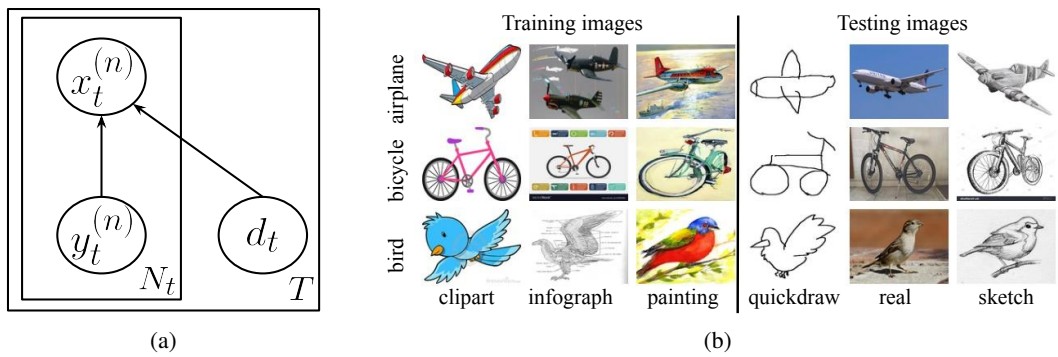

(a)                                                    (b)

Figure 1: (a) A causal model for the online continual learning under conditional domain shift. $T$ is the number of tasks and $N_t$ is the number of training samples of task $t$. The causal interaction between the domains and images are not considered in the traditional setting. (b) An example of the domain shift, sampled images are extracted from the DomainNet dataset (Peng et al., 2019).

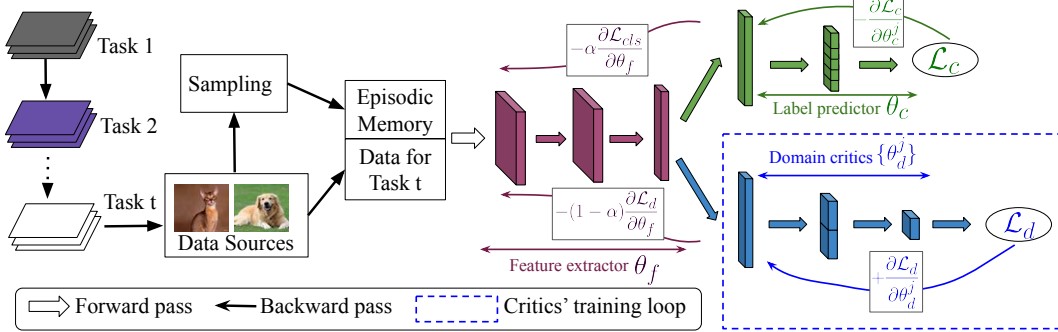

Figure 2: Overview of Conditional Invariant Experience Replay (CIER). CIER employs a set of domain critics (blue) to learn a conditional invariant representation via an adversarial training loss in every experience replay round. Best viewed in colors.

object. As a result, the model needs to achieve an invariant representation using data from various source domains observed during training, i.e. the joint distribution $P(X, Y)$ is the same for all source domains. To this end, we develop *Conditional Invariant Experience Replay (CIER)* that can simultaneously retain previous knowledge, learn new tasks, and generalize to new domains. CIER employs an episodic memory to perform experience replay and an adversarial loss to achieve an invariant representation. Particularly, CIER formulates a multiplayer minimax game such that the conditional distribution $P(X|Y)$ is stable across each class's observed source domains. Therefore, if the prior distribution $P(Y)$ is stable in the target domains, i.e., no class imbalance, CIER can achieve the invariant representation in the joint distribution $P(X, Y)$ and generalize to unseen domains. Fig. 2 provides a high level overview of the proposed CIER method.

In summary, we formalize the continual learning under conditional domain shift problem and construct *three* novel benchmarks using real data with different levels of domain shift and diversity in the number of domains, tasks, and classes. Then, we develop CIER, a novel continual learning method that learns a conditional invariant representation and generalizes to novel domains. Our extensive experiments demonstrate the limitations of existing continual learning methods when being tested on unseen domains and show that the proposed CIER can mitigate such domain gaps effectively.

## 2 RELATED WORK

### 2.1 CONTINUAL LEARNING

Continual learning aims at developing a model that can continuously learn different tasks over a data continuum. In literature, there are different continual learning protocols with different properties of the continuum. First, a setting can be either *task-free* (Rebuffi et al., 2017; Aljundi et al., 2019b) or *task-aware* (Kirkpatrick et al., 2017; Lopez-Paz & Ranzato, 2017) based on whether a task-indicator

is presented to the model. Second, learning can follow the *online* (Lopez-Paz & Ranzato, 2017) or *batch* (Kirkpatrick et al., 2017) setting. In online continual learning, the data within each task is also online and the model is only allowed to learn each example once. Differently, in batch continual learning, the model is allowed to learn a task over many epochs. Generally, the task-free and online settings are more difficult because the model has to predict on all tasks, and optimizing deep neural network online over one epoch is particularly challenging (Lopez-Paz & Ranzato, 2017; Sahoo et al., 2018; Aljundi et al., 2019a). However, we argue that existing protocols do not consider the domain shift between training and testing data of each task, which is important in real scenarios. This limitation motivates us in developing the continual learning under conditional domain shift problem as a practical and more challenging problem to evaluate continual learning methods.

Continual learning methods aim at achieving a good trade-off between alleviating catastrophic forgetting and acquiring new knowledge. Existing methods can be broadly categorized into two main groups: (i) dynamic architecture approaches that employ a separate subnetwork for each task, and (ii) static architecture approaches that maintain a fixed network throughout learning. While dynamic architecture methods (Rusu et al., 2016; Serra et al., 2018; von Oswald et al., 2020; Yoon et al., 2018; Li et al., 2019; Xu & Zhu, 2018) usually do not suffer from catastrophic forgetting, they are not practical because of the high computational cost. In the fixed architecture method family, experience replay (ER) (Lin, 1992) and its variants (Chaudhry et al., 2019b; Aljundi et al., 2019a; Riemer et al., 2019; Rolnick et al., 2019) have recently gained much interest thanks to their strong performance across different continual learning protocols, significantly outperforms other approachecs such as regularization (Kirkpatrick et al., 2017; Zenke et al., 2017; Aljundi et al., 2018; Ritter et al., 2018), representation learning (Rebuffi et al., 2017). However, existing continual learning methods lack the ability to generalize to unseen domains beyond training, which motivates us to develop a novel method of CIER that can achieve an invariant representation in continual learning.

## 2.2 DOMAIN GENERALIZATION AND DOMAIN ADAPTATION

Domain generalization aims at training a model that can generalize to a target domain given data from the source domains (Khosla et al., 2012; Li et al., 2017; 2018; Carlucci et al., 2019). As a simpler problem setting, domain adaptation assumes unsupervised data from the target domain is also given during training (Long et al., 2015; Ganin & Lempitsky, 2015). A general approach for such problems is learning an invariant representation by correcting the shift in $P(X, Y)$ across domains. Particularly, (Long et al., 2015; Ganin & Lempitsky, 2015) assume that the conditional distribution $P(Y|X)$ is the same across domains; therefore, the domain shift can be addressed by matching only $P(X)$. However, invariant in $P(Y|X)$ is a strong assumption and is unlikely to hold in many applications. A more robust approach is matching the conditional distribution $P(X|Y)$ (Zhang et al., 2013; Li et al., 2018). In which case, if the class-prior distribution $P(Y)$ is stable, i.e., there is no class-imbalance among domains, matching $P(X|Y)$ will result in the invariant in $P(X, Y)$ because of the decomposition $P(X, Y) = P(Y)P(X|Y)$. We emphasize that no imbalance data is a mild assumption, especially when we do not know about the target domains. This result motivated us in the development of CIER. However, different from such methods, CIER can accommodate novel classes and domains over time while maintaining good performances over all classes.

## 3 PROBLEM FORMULATION

### 3.1 ONLINE CONTINUAL LEARNING UNDER CONDITIONAL DOMAIN SHIFT

**Notations.** Let $\mathcal{X} \subset \mathbb{R}^D$ and $\mathcal{Y} \subset \mathbb{N}$ be the input and label space respectively. A domain is defined as a joint distribution $P(X, Y)$ over $\mathcal{X} \times \mathcal{Y}$, and the $k$-th domain $P^k(X, Y)$ is denoted as $P^k$. Often, we do not have access to the domain $P^k$, instead, we can label each data point with a scalar $d \in \mathbb{N}$ indicating that it comes from domain $d$. During training, a learner $\boldsymbol{\theta}$ receives a continuum of data $\{\boldsymbol{x}_i, d_i, y_i\}$, where $(\boldsymbol{x}_i, y_i)$ is the $i$-th training sample and $d_i$ is the domain indicator associated with $\{\boldsymbol{x}_i, y_i\}$. In the following, we use $y(t)$ as the number of observed classes until time $t$; $d_y(t)$ denotes the number of observed domains of class $y$ until time $t$; and $\boldsymbol{x} \sim Y_j^k$ denotes the samples of the $j$-th class and has domain $d_k$.

We consider the task-free, online continual learning setting where the continuum is *locally iid*, i.e. each training sample $(\boldsymbol{x}_i, y_i)$ is drawn iid from an underlying task distribution $P_{t_i}(X, Y)$ with the task identifier $t_i$ not revealed to the learner. Moreover, each task consists of a subset of labels sam-

pled without replacement from the label space $\mathcal{Y}$. To formalize the online continual learning under conditional domain shift problem, we assume that each class $y$ is associated with a set of source domains $\mathcal{D}_y^{src} = \{d_1^{src}, d_2^{src}, \ldots, d_s^{src}\}$, and a set of target domains $\mathcal{D}_y^{tgt} = \{d_1^{tgt}, d_2^{tgt}, \ldots, d_m^{tgt}\}$. Additionally, the source and target domains are disjoint for all classes, i.e. $\mathcal{D}_y^{src} \cap \mathcal{D}_y^{tgt} = \emptyset, \forall y$. We consider *three* scenarios with decreasing difficulties to evaluate the learner $\boldsymbol{\theta}$ based on the degree of shift between the training and testing domains.

- **Scenario 1** (significant shift): Train on source domains, test on target domains, the most challenging setting where all domains of the test data are unobserved during training.

- **Scenario 2** (mild shift): Train on source domains, test on both source and target domains. Here we assume that domains of test data can be draw from both source and target domains.

- **Scenario 3** (no shift): Train on target domains, test on target domains. This is the traditional continual learning setting where there is no discrepancy among domains of training and testing data. We design this scenario so that it has the same test data as the first one.

In all scenarios, the model is required to perform well on the test data. We emphasize that the continual learning setting we considered is the most challenging in the literature: both task and data arrive online, and the model has to predict all observed classes so far during evaluation. Moreover, we provide the model with the domain identifiers during training; however, only the input $x$ is provided at test time. Finally, we aim at creating a model that can predict at any moment, therefore, we do not make any assumption regarding the testing domains, including the availability of unlabeled data, and the models are not allowed to train during inference.

## 3.2 Evaluation Protocol

We use three standard metrics to measure the model's performance: Average Accuracy (Lopez-Paz & Ranzato, 2017), Forgetting Measure (Chaudhry et al., 2019a), and Learning Accuracy (Riemer et al., 2019). Let $T$ be the total number of tasks, $a_{i,j}$ be the model's accuracy evaluated on the test set of task $\mathcal{T}_j$ after it finished learning the task $\mathcal{T}_i$. The aforementioned metrics are defined as:

- **Average Accuracy** (higher is better): $\text{ACC} = \frac{1}{T} \sum_{i=1}^{T} a_{T,i}$.

- **Forgetting Measure** (lower is better): $\text{FM} = \frac{1}{T-1} \sum_{j=1}^{T-1} \max_{l \in \{1, \ldots T-1\}} a_{l,j} - a_{T,j}$.

- **Learning Accuracy** (higher is better): $\text{LA} = \frac{1}{T} \sum_{i=1}^{T} a_{i,i}$.

The Averaged Accuracy (ACC) measures the model's overall performance across all tasks and is a common metric to compare among different methods. Forgetting Measure (FM) measures the model's forgetting as the averaged performance degrades of old tasks. Finally, Learning Accuracy (LA) measures the model's ability to acquire new knowledge. Note that the task identifiers are *not* given to the model at any time and are only used to measure the evaluation metrics.

## 4 Proposed Method

### 4.1 Conditional Invariant As A Multiplayer Minimax Game

Learning an invariant representation is equivalent to correcting the shift in the joint distribution $P(X, Y) = P(Y)P(X|Y)$. However, it is often expensive to obtain the join distribution; therefore, we propose to learn a conditional invariant in the conditional distribution $P(X|Y)$. Then if the class-prior distributions $P(Y)$ satisfy $P^m(Y) = P^l(Y), \forall m \in \mathcal{D}_y^{src}, \forall l \in \mathcal{D}_y^{tgt}$, we can expect the model to generalize to unseen target domains. Therefore, we propose to learn a conditional invariant deep network $F_{\boldsymbol{\theta}}(\cdot)$ using only data from source domains by enforcing $P^k(F(X)|Y) = P^l(F(X)|Y), \forall k, l \in \mathcal{D}_y^{src}$.

To achieve this goal, we formulate learning a conditional invariant feature extractor as a multiplayer minimax game. Particularly, each class $y_j$ is associated with a domain critic $K_j$ predicting the domain associated with each sample of $y_j$. Therefore, we propose the following minimax problem:

$$\min_F \max_{\boldsymbol{\theta}_d^j, \ldots, \boldsymbol{\theta}_d^{y(t)}} \sum_{j=1}^{y(t)} V_{cond}(F, K_j), \tag{1}$$

where $V_{cond}(F, K_j)$ is a domain discrepancy measure for class $y_j$. At time $t$, Eq. 1 is a $(y(t) + 1)$–players minimax game with $y(t)$ critics trained to distinguish the features generated from the feature extractor $F$, while $F$ is trained simultaneously maximize all critics' losses. In this work, we consider two versions of the domain discrepancy measure $V_{cond}$. First, we consider the Jensen-Shannon divergence (JSD), in which case the domain discrepancy is calculated as $V_{cond}(F, K_j) = \sum_{i=1}^{m_j} \mathbb{E}_{\boldsymbol{x} \sim Y_j} \log K_j(F(\boldsymbol{x}))$. When using JSD, the domain critics $K_j$ are implemented as a softmax layer, i.e. $\sum_{i=1}^{d_j(t)} K_j(z) = 1, \forall j = 1, \ldots, y(t)$. JSD is a natural choice as the domain divergence as has been successfully applied in other applications, e.g. (Goodfellow et al., 2014; Li et al., 2018).

Another popular choice to measure domain divergence is the Wasserstein distance between two distributions (Villani, 2008). Different from JSD, the Wasserstein metric has a much smoother value range, which may provide more meaningful gradients to update the feature extractor. This motivates us to use the Wasserstein distance as the domain divergence in the minimax game in Eq. 1. In practice, estimating the Wasserstein-p distance for an arbitrary value of p is highly computationally expensive. Fortunately, the Kantorovich-Rubinstein duality (Villani, 2008) provides an efficient way to compute the Wasserstein-1 distance, which we refer to as the Wasserstein distance throughout the rest of this paper. The duality of the Wasserstein-1 distance can be efficiently computed as:

$$W(\mu, \nu) = \sup_{||f||_L \leq 1} \mathbb{E}_{X \sim \mu} f(X) - \mathbb{E}_{X \sim \nu} f(X), \tag{2}$$

where the supremum is taken over all Lipschitz continuous functions $f : M \to \mathbb{R}$ with Lipschitz constant less than or equal to one. The total domain divergence of class $y_j$ is calculated by summing the Wasserstein distances of all possible source domain pairs:

$$W_{cond}(F, K_j) = \sup_{||K_j||_L \leq 1} \left( \sum_{l \neq k} \mathbb{E}_{\boldsymbol{x} \sim Y_j^k} K_j(F(\boldsymbol{x})) - \mathbb{E}_{\boldsymbol{x} \sim Y_j^l} K_j(F(\boldsymbol{x})) \right) \tag{3}$$

Estimating the Wasserstein distance in Eq. 3 requires taking the supremum over all 1-Lipschitz functions of the critics. Here we follow the strategy of Arjovsky et al. (2017) to enforce this constraint by clipping the critics' parameters to be in a pre-defined range.

## 4.2 Conditional Invariant Experience Replay (CIER)

We now describe Conditional Invariant Experience Replay (CIER), which learns a conditional invariant feature representation of the data for continual learning. CIER utilizes an episodic memory to store and replay past data during learning new samples by minimizing the classification error and learning the conditional invariant features. To achieve the conditional invariant property, we propose to optimize the minimax objective in Eq. 1 each time a new sample arrives. Moreover, because previous data might not be fully available when learning new tasks, we restrict Eq. 1 to consider only data in the episodic memory. Therefore, CIER always learns conditional invariant features of all classes and source domains observed so far. To formalize CIER, let $\{\boldsymbol{\theta}_d^i\}_{i=1}^{y(t)}$ be the parameters of $y(t)$ domain critics $K_i$, $\boldsymbol{\theta}_f$ be the feature extractor, and $\boldsymbol{\theta}_c$ be the classifier. Then, the classification loss $L_{cls}(\boldsymbol{\theta}_f, \boldsymbol{\theta}_c)$ is the standard cross-entropy between the true and predicted labels. We propose two variants of CIER: CIER-JSD using the JSD, and CIER-W using Wasserstein as the domain divergence. The critics' losses $\mathcal{L}_d$ can be empirically estimated as

$$JSD : \mathcal{L}_d \equiv L_{JSD}(\boldsymbol{\theta}_f, \{\boldsymbol{\theta}_d^j\}) = \sum_j \left( \frac{1}{N_j^k} \sum_{\boldsymbol{x} \sim Y_j} \log K_{\boldsymbol{\theta}_d^j}(F_{\boldsymbol{\theta}_f}(\boldsymbol{x})) \right) ; \quad \sum_{i=1}^{d_j(t)} K_{\boldsymbol{\theta}_d^j}(z) = 1, \tag{4}$$

$$W : \mathcal{L}_d \equiv L_W(\boldsymbol{\theta}_f, \{\boldsymbol{\theta}_d^j\}) = \sum_j \left( \sum_{l \neq k} \frac{1}{N_j^k} \sum_{\boldsymbol{x} \sim Y_j^k} K_{\boldsymbol{\theta}_d^j}(F_{\boldsymbol{\theta}_f}(\boldsymbol{x})) - \frac{1}{N_j^k} \sum_{\boldsymbol{x} \sim Y_j^l} K_{\boldsymbol{\theta}_d^j}(F_{\boldsymbol{\theta}_f}(\boldsymbol{x})) \right), \tag{5}$$

where $N_j^k$ denotes the number of samples from class $j$ that belong to domain $k$ stored in the episodic memory. Let $L_{total} = \alpha L_{cls}(\boldsymbol{\theta}_f, \boldsymbol{\theta}_c) + (1 - \alpha) L_d(\boldsymbol{\theta}_f, \{\boldsymbol{\theta}_d^j\})$ be the total loss with the trade-off parameter $\alpha$. When an incoming data arrive, the whole network can be optimized by a two-stages

Table 1: Summary of the benchmarks used in our experiments

|  | #src domains | #tgt domains | #total domains | #train imgs | #test imgs | #classes |
|---|---|---|---|---|---|---|
| iAnimal | 23 | 10 | 33 | 4,500 | 1,500 | 10 |
| iVehicle | 28 | 11 | 39 | 4,000 | 1,290 | 9 |
| iDomainNet | 3 | 3 | 6 | 205,141 | 88,204 | 345 |

optimization process. First, the critics are trained to **maximize** the critic loss in Eq. 4, or Eq. 5. In the second stage, the classifier and the feature extractor are trained to correctly classify the current data as well as learning conditional invariant features. Importantly, the feature extractor $\theta_f$ is trained to minimize the domain's discrepancy measure while the critics are trained to maximize it, which implements the minimax game to learn conditional invariant features described in Sec. 4.1. Notably, calculating the critics loss $\mathcal{L}_d$ requires $y(t)$ forward passes, however, only one backward pass is required to update the critics' parameters. We provide the pseudo-code of both CIER-JSD and CIER-W in Appendix A.

## 5 EXPERIMENT

### 5.1 EXPERIMENTAL SETUP

**Benchmark.** We consider the task-free, online continual learning setting as described in Sec. 3: the model observes a continuum of data consists of $T$ tasks, where the input of each time step is a tuple of an image, domain indicator, and label. At test time, the model only observes the images $x$, and it has to predict on *all observed classes* so far. We consider two datasets: NICO (He et al., 2020) and DomainNet (Peng et al., 2019), and create three continual learning benchmarks in our experiments. The NICO dataset consists of two superclasses of Vehicle and Animal with 9 and 10 categories; each has a total of approximately 10,000 images over 30 domains. We create a continual learning benchmark for each superclass using its subclasses, resulting in the incremental Vehicle (iVehicle) and incremental Animal (iAnimal) benchmarks. Similarly, we take DomainNet, a large scale dataset of 345 classes with 569,010 images over six domains, and create the iDomainNet benchmark.

To ensure the domain shift between training and testing data, for each class, we randomly choose 50% of its domains as the source domains for training; the remaining domains are considered as target domains. The source and target domains are used to construct three continual learning scenarios as described in Sec. 3: significant shift, mild shift, and no shift. All images are resized, center-cropped to $3 \times 84 \times 84$ and normalized to be in $[0, 1]$. No other data augmentation is used in all experiments. Table 1 summaries the statistic of each benchmark. We can see that while the NICO dataset consists of many domains, it only has a small number of classes and a moderate amount of training data. On the other hand, the DomainNet dataset has a large number of classes and training images but only contains six domains. Consequently, our continual learning benchmarks are diverse in both number of training instances, domains, and tasks.

**Baselines.** Since there have not been any approaches in the literature that directly address the continual learning under domain shift problems, we consider three state-of-the-art online continual learning methods, namely ER (Chaudhry et al., 2019b), MER (Riemer et al., 2019), and MIR (Aljundi et al., 2019a). Furthermore, we also propose a baseline called Adversarial Experience Replay (AER) that augments ER with an adversarial domain classification loss to match the marginal distribution $P(X)$. AER is principally different from our proposed CIER since CIER matches the conditional distributions $P(X|Y)$ across domains. Due to space constraints, we refer to our Appendix B for the results of more baselines.

**Implementation Details.** We use a pre-trained ResNet18 (He et al., 2016) backbone in all benchmark. All methods use a Reservoir sampling buffer with sizes equal to 9,000; 10,000; and 34,500 samples in the iVehicle, iAnimal, and iDomainNet respectively. All methods are optimized online (over one epoch) using SGD with mini-batches of size 10 on iVehicle, iAnimal, and 32 on iDomainNet. We cross-validate the hyper-parameters of all methods on a small subset of data (5%) using grid search. Different from (Aljundi et al., 2019a), we follow the standard implementation in (Chaudhry et al., 2019b), which concatenates the current and previous data when calculating the experience replay loss. This results in the batch-normalization layer applied on both old and new data in the forward pass instead of the old and new data are batch-normalized separately. We observe that

Table 2: Evaluation metrics on the iVehicle and iAnimal benchmarks, respectively. All methods use the same pre-trained Resnet18 backbone. Scenario 1 (significant shift): train on source domains, test on target domains. Scenario 2 (mild shift): train on source domains, test on source and target domains. Scenario 3 (no shift): train on target domains, test on target domains.

| | iVehicle - 3 tasks | | | iAnimal - 5 tasks | | |
|---|---|---|---|---|---|---|
| Scenario 1 | ACC | FM | LA | ACC | FM | LA |
| MER | 48.12±1.12 | 22.58±2.56 | 63.20±1.43 | 46.38±1.00 | 31.50±2.05 | 71.62±1.41 |
| ER | 48.70±0.90 | 23.22±3.15 | 64.26±1.31 | 45.76±0.88 | 34.12±1.46 | 73.08±1.64 |
| MIR | 45.88±2.00 | 30.00±2.28 | 65.88±1.54 | 44.02±0.91 | 41.96±1.55 | **77.56±1.22** |
| AER | 48.96±0.39 | 24.30±1.32 | 64.46±0.98 | 45.98±1.47 | 32.82±2.02 | 72.24±0.99 |
| CIER-JSD | **51.52±0.59** | **19.62±1.98** | 65.78±1.28 | **49.04±0.97** | **31.20±1.16** | 74.00±1.53 |
| CIER-W | **51.38±1.10** | 22.04±2.63 | **66.08±1.33** | 48.36±0.67 | **31.02±1.77** | 73.16±1.57 |
| Scenario 2 | ACC | FM | LA | ACC | FM | LA |
| MER | 56.90±1.05 | 18.90±1.40 | 69.22±1.21 | 52.66±1.53 | **28.28±1.26** | 75.30±1.25 |
| ER | 56.30±1.29 | 21.04±2.44 | 70.34±1.20 | 51.60±0.80 | 31.68±1.52 | 76.92±1.74 |
| MIR | 55.46±0.88 | 25.24±1.34 | **72.30±0.34** | 49.16±1.52 | 40.52±1.32 | **81.58±0.95** |
| AER | 55.14±1.04 | 22.78±1.60 | 70.32±1.32 | 49.98±0.89 | 33.10±1.18 | 76.44±0.60 |
| CIER-JSD | **59.44±0.82** | **18.82±2.80** | 71.98±1.81 | 53.86±0.72 | 31.30±1.46 | 78.90±1.05 |
| CIER-W | 58.20±1.21 | 20.56±2.26 | 71.92±1.31 | **54.66±2.51** | 29.08±3.73 | 77.92±0.75 |
| Scenario 3 | ACC | FM | LA | ACC | FM | LA |
| MER | 63.14±0.84 | **20.28±1.61** | 76.70±0.88 | 61.74±0.62 | 26.42±1.14 | 82.88±0.46 |
| ER | 61.80±0.94 | 24.40±1.94 | 78.10±1.07 | 59.32±0.91 | 30.20±0.80 | 83.48±0.74 |
| MIR | 59.08±0.25 | 30.98±2.33 | 79.76±2.33 | 58.32±1.31 | 35.68±1.26 | **86.88±0.94** |
| AER | 62.82±0.87 | 20.64±2.50 | 76.56±1.26 | 59.52±0.53 | 28.84±0.70 | 82.56±0.97 |
| CIER-JSD | **65.78±1.28** | 23.62±2.06 | **81.54±0.80** | **63.78±1.13** | **26.10±1.71** | 83.56±2.06 |
| CIER-W | 64.82±1.28 | 24.10±1.33 | 80.90±0.61 | 62.32±0.80 | 29.30±0.90 | 85.68±1.20 |

concatenating current and episodic memory during for experience replay yields better results for **all methods**; therefore, we will use this implementation in the main paper. We provide the results of the other implementation with batch-normalization in the Appendix B.2.

## 5.2 RESULTS ON THE iVEHICLE AND iANIMAL BENCHMARKS

We create a sequence of three tasks for the iVehicle and five tasks for the iAnimal benchmarks and report the results in Table 2. Because of the standard ER implementation with batch-normalization, MIR performs slightly worse than ER across all settings. Under the significant domain shift, the performance of existing methods drops significantly compared to the traditional setting with no domain shift. Both ER, AER, CIER-JSD, and CIER-W has similar FM, indicating that they are equally affected by catastrophic forgetting. Moreover, CIER-JSD and CIER-W achieve higher LA, resulting in higher overall performance, indicated by the ACC metric. The results show that CIER can achieve a good trade-off between preventing catastrophic forgetting and facilitating knowledge transfer while being able to generalize to unseen domains.

## 5.3 RESULTS ON THE iDOMAINNET BENCHMARK

We construct a sequence of five and 15 tasks for the iDomainNet benchmark and report the results in Table 3. We also observe a significant performance drop across all continual learning methods. Moreover, since iDomainNet only has three source domains, it is much more challenging to learn an invariant representation. Consequently, we observe a slightly small gap between CIER and ER compared to the iVehicle and iAnimal benchmarks: 1% versus 2-4%. This result suggests that invariant representation learning requires more source domains to obtain a powerful representation that can generalize well at test time. Different from iAnimal and iVehicle, CIER-W consistently achieves the best performance in iDomainNet, showing that the Wasserstein distance is a better choice for the domain divergence when there are only a small number of source domains. When there are many source domains, it is challenging to extend the Wasserstein distance to multiple distributions while the JSD can be naturally extended to such cases.

Table 3: Evaluation metrics on the iDomainNet benchmarks with five and 15 tasks, respectively. All methods use the same pre-trained Resnet18 backbone. Scenario 1 (significant shift): train on source domains, test on target domains. Scenario 2 (mild shift): train on source domains, test on source and target domains. Scenario 3 (no shift): train on target domains, test on target domains

| | iDomainNet - 5 tasks | | | iDomainNet - 15 tasks | | |
|---|---|---|---|---|---|---|
| Scenario 1 | ACC | FM | LA | ACC | FM | LA |
| MER | 13.32±0.15 | **9.66±0.23** | 20.92±0.24 | 12.52±0.21 | **21.74±0.21** | 32.82±0.18 |
| ER | 15.06±0.18 | 9.88±0.52 | 22.96±0.27 | 12.60±0.22 | 22.56±0.23 | 34.62±0.24 |
| MIR | 15.18±0.30 | 10.26±0.29 | 23.40±0.16 | 12.28±0.27 | 23.14±0.13 | 34.88±0.30 |
| AER | 15.04±0.17 | 9.76±0.24 | 22.88±0.19 | 12.30±0.20 | 22.20±0.35 | 33.02±0.36 |
| CIER-JSD | 15.88±0.20 | 10.52±0.10 | 24.26±0.23 | 13.34±0.09 | 22.28±0.30 | 34.90±0.24 |
| CIER-W | **16.02±0.15** | 10.72±0.39 | **24.66±0.29** | **13.54±0.19** | 22.44±0.22 | **35.18±0.10** |
| Scenario 2 | ACC | FM | LA | ACC | FM | LA |
| MER | 23.96±0.09 | 13.06±0.13 | 34.42±0.19 | 23.28±0.15 | **25.98±0.21** | 47.24±0.12 |
| ER | 26.96±0.09 | 12.50±0.13 | 36.92±0.12 | 23.98±0.21 | 25.90±0.24 | 49.16±0.19 |
| MIR | 26.88±0.18 | 12.68±0.24 | 37.04±0.18 | 23.80±0.10 | 26.72±0.10 | 49.70±0.10 |
| AER | 26.62±0.90 | **12.08±0.90** | 36.28±0.13 | 22.68±0.10 | 26.52±0.13 | 47.44±0.04 |
| CIER-JSD | 27.96±0.17 | 13.14±0.27 | 38.48±0.07 | 24.74±0.04 | 26.12±0.09 | 49.82±0.07 |
| CIER-W | **28.16±0.18** | 13.24±0.36 | **38.74±0.19** | **24.92±0.22** | 26.04±0.29 | **49.92±0.09** |
| Scenario 3 | ACC | FM | LA | ACC | FM | LA |
| MER | 39.06±0.11 | **15.96±0.22** | 51.64±0.24 | 37.30±0.18 | **32.46±0.23** | 67.58±0.10 |
| ER | 41.12±0.07 | 17.06±0.12 | 54.80±0.14 | 38.42±0.23 | 33.14±0.30 | 70.34±0.09 |
| MIR | 40.88±0.14 | 17.66±0.20 | 55.00±0.10 | 37.84±0.41 | 34.10±0.48 | 70.64±0.09 |
| AER | 40.62±0.04 | 17.28±0.31 | 54.44±0.24 | 34.24±0.13 | 35.62±0.13 | 67.50±0.15 |
| CIER-JSD | 43.74±0.17 | 17.68±0.25 | 57.86±0.17 | 38.62±0.32 | 33.66±0.33 | 70.44±0.04 |
| CIER-W | **43.94±0.70** | 17.78±0.10 | **58.16±0.90** | **39.12±0.21** | 33.34±0.26 | **71.04±0.13** |

Finally, we note that all methods have lower ACC when moving from 5 to 15 tasks. Since both cases have the same amount of training data; having more tasks means each task is easier to learn, resulting in higher LA. However, learning on a long sequence of tasks also suffers significantly from catastrophic forgetting indicated by higher FM, resulting in lower final performances.

## 5.4 MODEL AND COMPUTATIONAL COMPLEXITY

Table 4: Model and computational complexity of CIER compared to ER and MIR. Computational complexity is the averaged training time per incoming batch, all methods have very small stddev, i.e., smaller than 0.01s

| | ER | | MIR | | CIER-JSD | | CIER-W | |
|---|---|---|---|---|---|---|---|---|
| | # params | Time | # params | Time | # params | Time | # params | Time |
| iVehicle | 11,181,129 | 0.42s | 11,181,129 | 0.53s | 11,443,013 | 0.53s | 11,380,562 | 0.48s |
| iAnimal | 11,181,642 | 0.42s | 11,181,642 | 0.54s | 11,437,872 | 0.54s | 11,381,332 | 0.48s |
| iDomainNet | 11,353,497 | 0.57s | 11,353,497 | 0.59s | 11,816,612 | 0.87s | 11,639,282 | 0.68s |

We study the model and computational complexity of CIER, and report the number of parameters and the averaged training time per incoming data on all benchmarks in Table 4. Among all methods, we observe significant additional computational cost in the iDomainNet benchmark. The overhead incurs because of the larger batch size and the larger number of classes: 345 classes compared to nine in iVhehicle and 10 in iAnimal. In CIER-JSD, each class's critic is a D-ways classification model where D is the number of domains observed, which results in more expensive parameters and computational complexity. On the other hand, CIER-W's critics only predict domains scores, which are scalars. Thefore, CIER-W enjoys much lower complexities compared to CIER-JSD. However,

the complexity overhead in both methods are not significant compared to ER and especially MIR. Particularly, CIER-W is only 15%-20% slower than ER, and has 1.7%-2.5% additional parameters.

## 5.5 SUMMARY

From the experiments, we observe that all continual learning methods suffer when a domain shift happens. Second, having many source domains is important in achieving an invariant representation, as demonstrated by the gap of CIER compared to other methods from the considered benchmarks. Moreover, the Wasserstein distance is powerful when learning with a small number of source domains while the Jensen-Shannon divergence (JSD) is a better choice when more source domains are available. Finally, while CIER-JSD and CIER-W achieved promising results, there is still a gap compared to the ideal scenario of having no domain shift between training and testing domains.

## 6 CONCLUSION

We address the limitations of existing continual learning methods when testing on unseen domains with domain shift. This motivates to develop a novel problem setting named Continual learning under domain shift and three novel benchmarks, which support a diverse range of domains, classes, and tasks. To address this challenge, we propose CIER that simultaneously performs experience replay to maintain a good performance across all tasks and learns a conditional invariant representation via an adversarial learning objective. As a result, CIER balances among all *three* aspects: preventing catastrophic forgetting, facilitating knowledge transfer, and generalization to unseen domains. Through extensive experiments, we demonstrate that existing state-of-the-art methods are often too optimistic and struggle to generalize to unseen domains. On the other hand, CIER can bridge the gap in the domain shifts and offer significant improvements. We believe our study provides a more realistic setting to evaluate continual learning methods and opens new future research directions.

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

This Appendix is organized as follows. We describe the pseudo-code of the CIER algorithms in Appendix A. Then, Appendix B provides the details of our implementation used in the main paper and additional results of more baselines and model complexity. Finally, we include some illustrative examples extracted from our benchmarks in Appendix C.

## A    THE CIER ALGORITHMS

Alg. 1 depicts the training procedure of the proposed CIER-JSD and CIER-W. All experiments conducted in this work use the hyperpameter provided in Alg. 1. For a fair comparisons, we optimize all methods, including CIER and other baselines, using two updates per samples.

---

**Algorithm 1:** Our proposed CIER algorithm. All experiments in this paper used the default values $n_{inner} = n_{critic} = 2$, $\alpha = 0.95$, $c = 0.01$.

---

**Require:** $\alpha$, the trade-off parameter. $c$, the clipping parameter. $N$, the replay batch size. $n_{iter}$: number of iterations per incoming training data.

**Init:** $\boldsymbol{\theta}_f$, the deep network feature extractor parameters. $\boldsymbol{\theta}_c$, the classifier parameters. $\{\boldsymbol{\theta}_d^i\}_{i=1}^{|Y|}$, the critics parameters of $|Y|$ classes. $\mathcal{M}$ the episodic memory with reservoir sampling.

1 **for** $t \leftarrow 1$ **to** $T$ **do**
2      **for** $j \leftarrow 1$ **to** $n_{batches}^t$ **do**         // Receive data of task $\mathcal{T}$ sequentially
3          $\mathcal{B}_j = (\boldsymbol{x}, d, y)$, receive a mini batch of data from the current task
4          **for** $m \leftarrow 1$ **to** $n_{inner}$ **do**         // Inner update of the current sample
5             $\mathcal{B}_{\mathcal{M}} = (\boldsymbol{x}', d', y')$ random sampling from the episodic memory $\mathcal{M}$
6             $\mathcal{B}_{\mathcal{M},m} \leftarrow \mathcal{B}_j \cup \mathcal{B}_{\mathcal{M}}$         // Mixing data for experience replay
7             **for** $n \leftarrow 0$ **to** $n_{critic}$ **do**         // Training critics loop
8                 Calculate the critics' loss $L_W$ on $\mathcal{B}_{\mathcal{M},m}$ using Eq. 4 or Eq. 5
9                 $\boldsymbol{\theta}_d^i \leftarrow \boldsymbol{\theta}_d^i + \gamma \frac{\partial L_d}{\partial \boldsymbol{\theta}_d^i}, \forall i = 1, \ldots, |Y|$         // Domain critics SGD step
10                 CIER-W: $\boldsymbol{\theta}_d^i \leftarrow \text{clip}(\boldsymbol{\theta}_d^i, -c, c), \forall i = 1, \ldots, |Y|$         // Parameters clipping
11             Calculate the total loss $L_{total}$ and the classification loss $L_{cls}$ on $\mathcal{B}_{\mathcal{M},m}$.
12             $\boldsymbol{\theta}_f \leftarrow \boldsymbol{\theta}_f - \gamma \frac{\partial L_{total}}{\partial \boldsymbol{\theta}_f}$         // Feature extractor SGD step
13             $\boldsymbol{\theta}_c \leftarrow \boldsymbol{\theta}_c - \gamma \frac{\partial L_{cls}}{\partial \boldsymbol{\theta}_c}$         // Classifier SGD step
14          Add $\mathcal{B}_j$ to $\mathcal{M}$ using reservoir sampling
15 **return** $\boldsymbol{\theta}_f, \{\boldsymbol{\theta}_d^i\}_{i=1}^{|Y|}, \boldsymbol{\theta}_c$

---

## B    EXPERIMENTS DETAILS

In this section, we first details the baselines used in our experiments. Then, we prove the details of experience replay implementations using batch normalization, and highlight the differences between them. For a comparison, we also include the results of both implementation on the iVehicle and iAnimal benchmarks. Finally, we provide the results of more continual learning methods on the iVehicle and iAnimal benchmarks.

### B.1    EXPERIENCE REPLAY IMPLEMENTATIONS WITH BATCH NORMALIZATION

We first describe two experience replay (ER) implementations considered in this work. Let $\mathcal{B}_t$ be the current data observed from the continuum and $\mathcal{B}_M$ be a random batch of data sampled from the episodic memory. For convenience, we use $\boldsymbol{\theta}$ to denote the total number of parameters in the model, including the feature extractor and the classifier. Then, an experience replay update can be implemented with or without concatenating $\mathcal{B}_t$ and $\mathcal{B}_M$ as followings.

**Batch Normalization with concatenation.** This is the standard implementation of ER used in (Chaudhry et al., 2019b). The update step is described as:

$$\bar{\mathcal{B}} \leftarrow \mathcal{B}_t \cup \mathcal{B}_M, \tag{6}$$

$$\mathcal{L}_{cls} \leftarrow \mathcal{L}(\boldsymbol{\theta}, \bar{\mathcal{B}}), \tag{7}$$

$$\boldsymbol{\theta} \leftarrow \text{SGD}(\mathcal{L}_{cls}, \boldsymbol{\theta}). \tag{8}$$

In this implementation, the batch normalization layer is applied on the merged batch $\bar{\boldsymbol{B}}$, which results in *one* forward pass, and the mini-batch mean and mini-batch variance being estimated once using the concatenation of the current data $\boldsymbol{B}_t$ and old data $\boldsymbol{B}_M$ in the memory.

**Batch Normalization without concatenation.** This is an alternative implementation of ER used in (Aljundi et al., 2019a). The update step is described as:

$$\mathcal{L}_{cls} \leftarrow \mathcal{L}(\boldsymbol{\theta}, \mathcal{B}_t) + \mathcal{L}(\boldsymbol{\theta}, \mathcal{B}_M), \tag{9}$$

$$\boldsymbol{\theta} \leftarrow \text{SGD}(\mathcal{L}_{cls}, \boldsymbol{\theta}). \tag{10}$$

In this implementation, the loss is calculate on current and previous data separately. This results in *two* forward passes, and the mini-batch mean and mini-batch variance being estimated twice per gradient update: once on the current data $\boldsymbol{B}_t$ and once on the past data $\boldsymbol{B}_M$.

Since ER is the base of many methods used in our work, including MIR (Aljundi et al., 2019a), our CIER-JSD and CIER-W, this results in two alternative implementations of such methods.

### B.2 RESULTS OF THE TWO EXPERIENCE REPLAY IMPLEMENTATIONS

We report the results of both experience replay implementations on the iVehicle in Table 5. We emphasize that in these implementations, beside the difference in concatenating the current and past samples, other factors are exactly the same. Interestingly, a subtle change in the use batch normalization can result in significant changes of the final results. One possible reason of this discrepancy is the use of the pre-trained ResNet, which was pre-trained by concatenating all samples in each mini-batch. In addition, when using BN without concatenation, we observe the same behaviour of MIR as in (Aljundi et al., 2019a): MIR improves ER by lowering forgetting measure FM, resulting in slightly better overall performance across benchmarks. Moreover, we still observe the same trend in both implementations that CIER can improve over ER and MIR in terms of LA, which results in better ACC. Nevertheless, implementing ER using BN with concatenation showed to to superior than the alternative. Therefore, we decided to report the result of this strategy in the main paper.

### B.3 RESULTS OF ADDITIONAL BASELINES

In this section, we provide a broader set of baselines than the ones included in the main paper. First, we provide a short description of all baselines considered.

- **Learning without Forgetting (LwF)** (Li & Hoiem, 2017). LwF prevents catastrophic forgetting by a variant of knowledge distillation.
- **AGEM** (Chaudhry et al., 2019a). AGEM prevents catastrophic forgetting and transferring knowledge by preventing the old tasks' losses from increasing via a quadratic constraint. Since the original AGEM experimented with the online, task-aware setting, we extend AGEM to the task-free setting by enforcing the constraint on a randomly selected memory data of all observed classes.
- **Meta Experience Replay (MER)** (Riemer et al., 2019). MER balances both catastrophic forgetting and facilitating knowledge transfer by penalizing the gradient direction of every samples stored in the episodic memory. MER implements this objective via a variant of the Reptile algorithm (Nichol et al., 2018).
- **Experience Replay (ER)** (Chaudhry et al., 2019b). ER is a simple, yet very effective strategy for the online, task-free continual learning problem. ER works by simply mixing current data and previous data in the memory and peforms a gradient update.
- **Maximally Interfered Retrieval ER (MIR)** (Aljundi et al., 2019a). MIR improves ER by replacing the random memory sampling with a strategy to selects previous samples causing the most classification loss increase.

Table 5: Evaluation metrics of two experience replay implementations on the iVehicle benchmarks. BN denotes Batch Normalization. Scenario 1: train on source domains, test on target domains. Scenario 2: train on source domains, test on source and target domains. Scenario 3: train on target domains, test on target domains

| iVehicle | BN with concatenation | | | BN without concatenation | | |
|---|---|---|---|---|---|---|
| Scenario 1 | ACC | FM | LA | ACC | FM | LA |
| ER | 48.70±0.90 | 23.22±3.15 | 64.26±1.31 | 41.70±1.10 | 20.12±4.10 | 55.04±2.95 |
| MIR | 45.88±2.00 | 30.00±2.28 | 65.88±1.54 | 41.96±1.25 | **20.04±2.01** | 55.30±1.48 |
| CIER-JSD | **51.52±0.59** | **19.62±1.98** | 65.78±1.28 | **43.26±1.81** | 20.66±3.29 | **57.02±2.34** |
| CIER-W | **51.38±1.10** | 22.04±2.63 | **66.08±1.33** | 42.69±2.45 | 20.54±4.65 | 55.70±0.83 |
| Scenario 2 | ACC | FM | LA | ACC | FM | LA |
| ER | 56.30±1.29 | 21.04±2.44 | 70.34±1.20 | 50.09±2.10 | 21.40±4.56 | 65.20±2.72 |
| MIR | 55.46±0.88 | 25.24±1.34 | **72.30±0.34** | 50.66±2.71 | **16.60±2.61** | 61.74±2.86 |
| CIER-JSD | **59.44±0.82** | **18.82±2.80** | 71.98±1.81 | **50.86±1.80** | 20.34±2.35 | **66.18±2.34** |
| CIER-W | 58.20±1.21 | 20.56±2.26 | 71.92±1.31 | 50.56±2.61 | 20.84±0.59 | 64.40±2.54 |
| Scenario 3 | ACC | FM | LA | ACC | FM | LA |
| ER | 61.80±0.94 | 24.40±1.94 | 78.10±1.07 | 58.48±1.04 | 16.92±2.94 | 69.78±2.39 |
| MIR | 59.08±0.25 | 30.98±2.33 | 79.76±2.33 | 59.34±0.50 | **13.70±2.36** | 68.48±1.22 |
| CIER-JSD | **65.78±1.28** | 23.62±2.06 | **81.54±0.80** | **60.16±0.76** | 15.58±1.54 | 70.52±1.58 |
| CIER-W | 64.82±1.28 | 24.10±1.33 | 80.90±0.61 | 59.84±1.23 | 16.90±3.61 | **71.48±2.54** |

Table 6: Evaluation metrics of two experience replay implementations on the iAnimal benchmarks. BN denotes Batch Normalization. Scenario 1: train on source domains, test on target domains. Scenario 2: train on source domains, test on source and target domains. Scenario 3: train on target domains, test on target domains

| iAnimal | BN with concatenation | | | BN without concatenation | | |
|---|---|---|---|---|---|---|
| Scenario 1 | ACC | FM | LA | ACC | FM | LA |
| ER | 45.76±0.88 | 34.12±1.46 | 73.08±1.64 | 41.84±1.88 | 29.60±3.65 | 65.38±1.88 |
| MIR | 44.02±0.91 | 41.96±1.55 | **77.56±1.22** | 42.96±3.24 | 28.72±4.22 | 66.48±1.33 |
| CIER-JSD | **49.04±0.97** | **31.20±1.16** | 74.00±1.53 | 43.52±0.41 | 29.69±0.61 | 67.48±2.41 |
| CIER-W | 48.36±0.67 | **31.02±1.77** | 73.16±1.57 | 42.99±1.33 | 31.76±1.99 | 67.66±0.99 |
| Scenario 2 | ACC | FM | LA | ACC | FM | LA |
| ER | 51.60±0.80 | 31.68±1.52 | 76.92±1.74 | 43.90±3.37 | 32.72±3.99 | 70.06±1.20 |
| MIR | 49.16±1.52 | 40.52±1.32 | **81.58±0.95** | 44.38±2.95 | 30.03±4.44 | 68.98±1.14 |
| CIER-JSD | 53.86±0.72 | 31.30±1.46 | 78.90±1.05 | 44.48±2.81 | 33.52±3.00 | 70.68±2.34 |
| CIER-W | **54.66±2.51** | 29.08±3.73 | 77.92±0.75 | 45.14±4.19 | 33.90±2.63 | 72.24±1.21 |
| Scenario 3 | ACC | FM | LA | ACC | FM | LA |
| ER | 59.32±0.91 | 30.20±0.80 | 83.48±0.74 | 55.14±1.08 | 25.46±1.50 | 75.42±1.07 |
| MIR | 58.32±1.31 | 35.68±1.26 | **86.88±0.94** | 56.02±1.11 | 22.62±2.22 | 74.78±1.77 |
| CIER-JSD | **63.78±1.13** | **26.10±1.71** | 83.56±2.06 | 56.48±2.31 | 24.08±2.69 | 73.94±0.95 |
| CIER-W | 62.32±0.80 | 29.30±0.90 | 85.68±1.20 | 56.16±2.26 | 27.08±1.93 | 77.54±1.40 |

- **Adversarial Experience Replay (AER)** (this work). AER improves over ER by matching the distribution $P(X)$ across domains. Particularly, AER employs a domain classifier to classify the domains from the backbone's features. The domain classifier is trained to maximize the domain classification loss while the feature extractor is trained to minimize it, resulting in an invariant in $P(X)$. As we discussed in Sec. 2, this method corresponds to an invariant assumption in $P(Y|X)$; however, when this assumption is violated, matching $P(X)$ may not provide additional benefits to the traditional ER.

Table 7: Evaluation metrics on the iVehicle and iAnimal benchmarks. All methods use the same pretrained Resnet18 backbone. Scenario 1: train on source domains, test on target domains. Scenario 2: train on source domains, test on source and target domains. Scenario 3: train on target domains, test on target domains. This is the full version of Table 2 in the main paper

| | iVehicle - 3 tasks | | | iAnimal - 5 tasks | | |
|---|---|---|---|---|---|---|
| Scenario 1 | ACC | FM | LA | ACC | FM | LA |
| LwF | 24.10±1.60 | 55.86±1.44 | 61.38±1.80 | 16.44±1.60 | 78.44±2.79 | 79.18±1.31 |
| AGEM | 23.76±0.43 | 69.68±1.32 | **70.18±1.19** | 15.34±1.01 | 85.96±2.05 | **84.14±1.25** |
| MER | 48.12±1.12 | 22.58±2.56 | 63.20±1.43 | 46.38±1.00 | 31.50±2.05 | 71.62±1.41 |
| ER | 48.70±0.90 | 23.22±3.15 | 64.26±1.31 | 45.76±0.88 | 34.12±1.46 | 73.08±1.64 |
| MIR | 45.88±2.00 | 30.00±2.28 | 65.88±1.54 | 44.02±0.91 | 41.96±1.55 | 77.56±1.22 |
| AER | 48.96±0.39 | 24.30±1.32 | 64.46±0.98 | 45.98±1.47 | 32.82±2.02 | 72.24±0.99 |
| CIER-JSD | **51.52±0.59** | **19.62±1.98** | 65.78±1.28 | **49.04±0.97** | **31.20±1.16** | 74.00±1.53 |
| CIER-W | **51.38±1.10** | 22.04±2.63 | 66.08±1.33 | 48.36±0.67 | **31.02±1.77** | 73.16±1.57 |
| Scenario 2 | ACC | FM | LA | ACC | FM | LA |
| LwF | 29.76±0.86 | 64.98±4.46 | 73.08±2.14 | 14.28±2.69 | 83.62±1.20 | 81.18±1.91 |
| AGEM | 25.86±0.65 | 77.18±0.99 | **77.34±1.20** | 15.16±0.90 | 88.34±1.01 | **85.80±1.10** |
| MER | 56.90±1.05 | 18.90±1.40 | 69.22±1.21 | 52.66±1.53 | **28.28±1.26** | 75.30±1.25 |
| ER | 56.30±1.29 | 21.04±2.44 | 70.34±1.20 | 51.60±0.80 | 31.68±1.52 | 76.92±1.74 |
| MIR | 55.46±0.88 | 25.24±1.34 | 72.30±0.34 | 49.16±1.52 | 40.52±1.32 | 81.58±0.95 |
| AER | 55.14±1.04 | 22.78±1.60 | 70.32±1.32 | 49.98±0.89 | 33.10±1.18 | 76.44±0.60 |
| CIER-JSD | **59.44±0.82** | **18.82±2.80** | 71.98±1.81 | 53.86±0.72 | 31.30±1.46 | 78.90±1.05 |
| CIER-W | 58.20±1.21 | 20.56±2.26 | 71.92±1.31 | **54.66±2.51** | 29.08±3.73 | 77.92±0.75 |
| Scenario 3 | ACC | FM | LA | ACC | FM | LA |
| LwF | 38.30±1.48 | 63.48±2.85 | 80.62±1.91 | 17.64±0.99 | 85.02±2.58 | 85.62±2.74 |
| AGEM | 27.88±1.70 | 81.52±1.59 | **82.24±0.95** | 16.60±0.57 | 91.88±0.38 | **90.10±0.86** |
| MER | 63.14±0.84 | **20.28±1.61** | 76.70±0.88 | 61.74±0.62 | 26.42±1.14 | 82.88±0.46 |
| ER | 61.80±0.94 | 24.40±1.94 | 78.10±1.07 | 59.32±0.91 | 30.20±0.80 | 83.48±0.74 |
| MIR | 59.08±0.25 | 30.98±2.33 | 79.76±2.33 | 58.32±1.31 | 35.68±1.26 | 86.88±0.94 |
| AER | 62.82±0.87 | 20.64±2.50 | 76.56±1.26 | 59.52±0.53 | 28.84±0.70 | 82.56±0.97 |
| CIER-JSD | **65.78±1.28** | 23.62±2.06 | 81.54±0.80 | **63.78±1.13** | **26.10±1.71** | 83.56±2.06 |
| CIER-W | 64.82±1.28 | 24.10±1.33 | 80.90±0.61 | 62.32±0.80 | 29.30±0.90 | 85.68±1.20 |

Due to the computational expensive of the iDomainNet benchmark, we only report the results of all baselines in the iVehicle and iAnimal benchmarks in Table 7. The results show that, in general, continual learning benchmarks cannot achieve a good balance among three important aspects: preventing catastrophic forgetting, facilitating knowledge transfer, and generalization to unseen domains. This is reflected by comparing the three evaluated metrics. While AGEM can facilitate knowledge transfer (high LA), it suffers from catastrophic forgetting (high FM), resulting in poor performances (low ACC). ON the other hand, experience replay and its variants can achieve a good trade-off between preventing catastrophic forgetting and facilitating knowledge transfer; but they struggle to generalize to unseen domain. Our result suggests that balancing among all three objectives is important for continual learning and remains an open problem for future research.

## C  SAMPLE IMAGES

We provide sample images of the iAnimal, iVehicle, and iDOmainNet benchmarks in Fig. 3, Fig. 4, and Fig. 5 respectively. For each benchmark, we show the sample images in the source and target domains of three randomly chosen classes. Each class's source and target domains are used to construct the three evaluation scenarios described in Sec. 3.

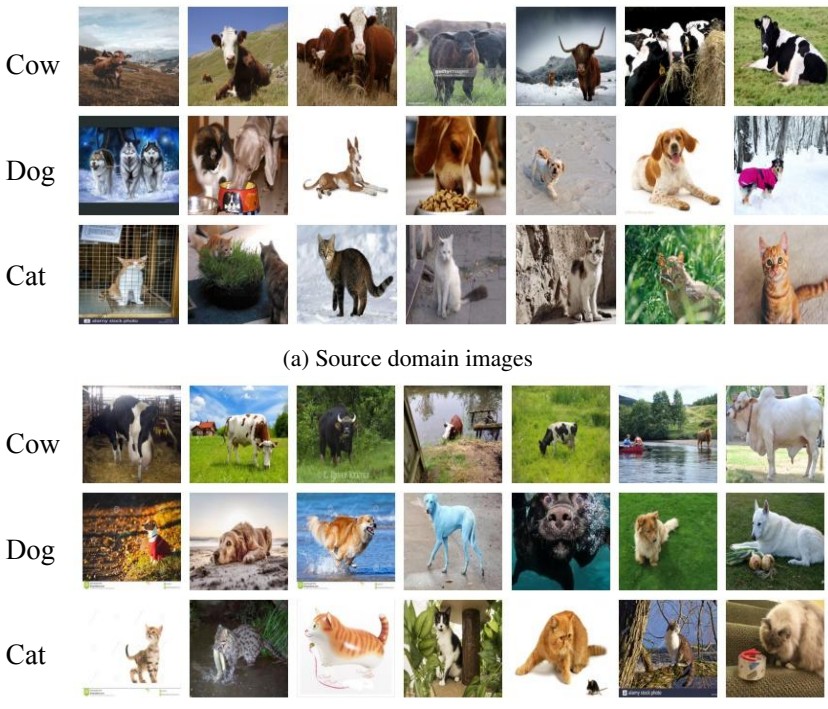

Figure 3: Sample images from the iANimal benchmark. Images are extracted from the Animal superclass of the NICO dataset (He et al., 2020).

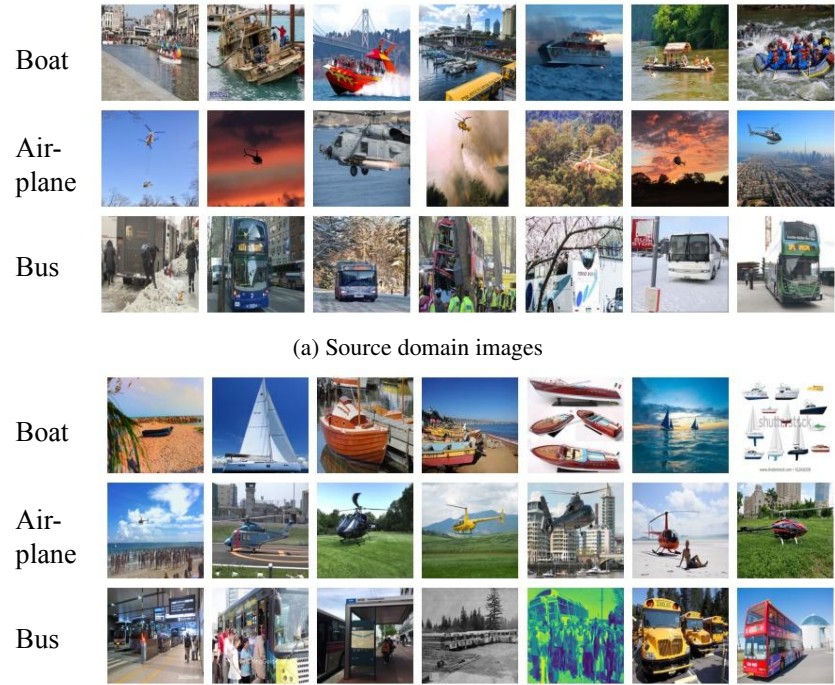

Figure 4: Sample images from the iVehicle benchmark. Images are extracted from the Vehicle superclass of the NICO dataset (He et al., 2020).

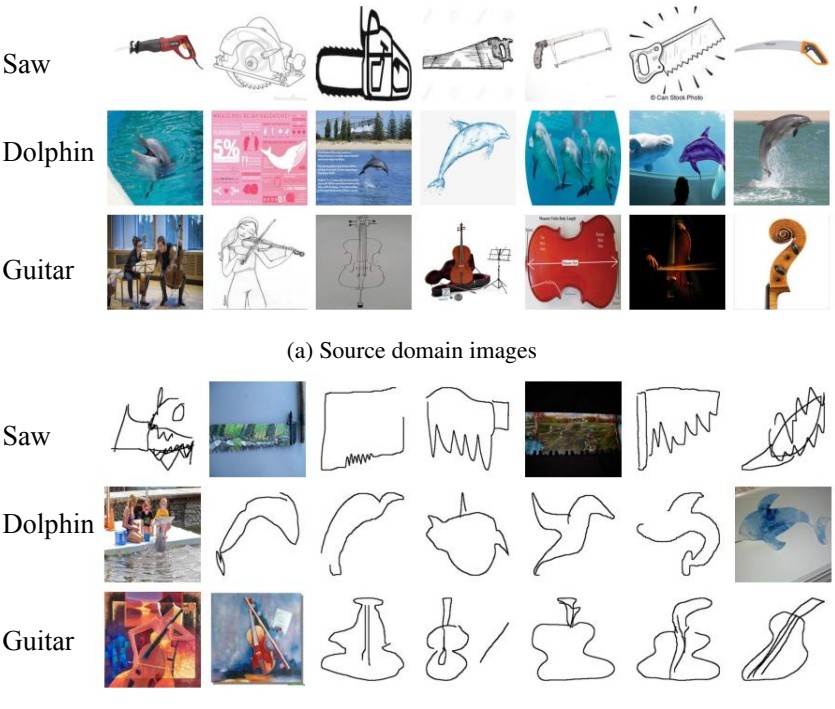

(a) Source domain images

(b) Target domain images

Figure 5: Sample images from the iDomainNet benchmark. Images are extracted from the Domain-Net dataset (Peng et al., 2019).

