# OpenReview forum: "Online Continual Learning  Under  Domain Shift"
_ICLR.cc/2021/Conference — Reject_

### Official Review · AnonReviewer3 · 2020-10-27
**Weak reject**

**Rating:** 5
**Confidence:** 3

**Review:**

I'd like to preface this review by saying that I am not an expert in this area of continual learning. Hence, I will adjust my review after reading the authors' response as well as other reviewers' comments accordingly.

This work proposes CIER, a continual learning method that adjusts to domain shift during test time. The authors claim that existing methods correct distribution shift in P(X, Y), which makes the stronger assumption that P(Y | X) is the same across domains. In contract, this work follow prior work by Zhang 2013 and corrects distribution shift in P(X | Y), which makes the weaker assumption that P(Y) is the same across domains. The contribution of this work include using an adversarial objective via a critic that attempts to distinguish the domain of the learned representation F(X | Y), as well as the adoption of an experience replay buffer from which to sample examples to optimize the minimax adversarial objective.

My concerns are as follows:
1. I am skeptical of the central claim of this work, which is that continual learning does not address domain shift. I am not an expert in this area, but to my knowledge, all reinforcement learning work that train on one environment and adapts to another environment must continual learn in a different environment (e.g. https://arxiv.org/abs/1803.11347, https://arxiv.org/abs/1905.04819, https://arxiv.org/abs/1910.08210). Can the authors comment on whether they think these works that both continually learn and adapt to new domains are relevant? If so, would it make sense to compare to them? Moreover, these works (and others in domain adaptation) do not assume that P(Y) is identical across domains, hence I feel like this is a rather strong assumption.
2. The manuscript rehash the story of learning an invariant F(X | Y) repeatedly, however the terms in which it does this are not precise. For example, how do the authors define: domain, task, test domain, target domain, invariance, stability? I suggest that the authors define these precisely (for example w/ mathematical definitions), and give concrete examples grounded in application settings where possible.
3. When the authors say that the distribution is stable, they mean that there is no class imbalance between domains. However stability in a distribution leads to a number of consequences (https://en.wikipedia.org/wiki/Stable_distribution) - are these consequences necessary? If so, in what ways? If not, would it make sense to just say "there is no class imbalance between domains"?
4. The manuscript contains some traces of notation abuse which make it hard to read. For example, D and its cursive variant represent critic and domain. the lowercase t represents both task and time, the authors alternate between test domain and target domain. I suggests that the authors make their terminology consistent.
5. In section 4.1, the authors propose learning a representation conditionally invariant in P(X|Y), however they also say that this was done by Zhang 2013 and Li 2018, hence this is not a contribution, correct?

In summary, due to the limitations posed by the assumption of this work (P(Y) the same across domains), the incremental contribution of this work (adversarial loss + experience replay), and the lack of clarity and precision in the manuscript,  I recommend a weak rejection. Due to my lack of experience in continual learning, I cannot assess the strength of the benchmarks and the significance of the experimental results, and will defer to other reviewers.

---

> ### Author Response · Authors · 2020-11-24
> **Response to Reviewer 3**
>
> We thank the Reviewer for your detailed and valuable feedback. Please find our response to your comments below.
>
> Concern #1: domain shift in existing works. We thank the Reviewer for mentioning related work in RL with domain shift. We want to clarify that our work focuses on the supervised learning setting, where the problem of continual learning under domain shift has not been extensively explored.  Moreover, there are certain differences in the problem setting making a direct comparison not possible, and extending these methods to supervised learning is not trivial. Particularly, Nagabandi et al, 2020 (https://arxiv.org/pdf/1803.11347.pdf) assumes an adaptation phase (online finetuning) at test time, which is unrealistic in many scenarios. For example, consider an image recognition system that has to be finetuned everytime it is required to make a prediction.
> In our domain adaptation formulation, we do not assume having access to any information about the testing domains, even the unlabeled data. Therefore, $P(Y)$ is the same between training and testing domain is considered as a condition in which the method can work. A more simplified formulation is the “unsupervised domain adaptation” setting, where during training, the model has access to some unlabeled data of the testing domains, which can be used to correct the changes in $P(Y)$. However, this is not the case in our work.
> We amend the draft accordingly to avoid possible confusion in the future.
>
> Concern #2: missing definitions. Thank you for your feedback. In the revision, we improved the notation definitions and provided sufficient descriptions throughout the paper, especially in the Notations paragraph, Section 3.1.
>
> Concern #3: stable distribution. Thank you for pointing out the confusion in our writing. Throughout this work, we wanted to emphasize that there was not class-imbalance between classes in training and testing domains.
>
> Concern #4: notation abuse. Thank you for your comment, in the revision we fixed the notation to improve the paper’s clarity.
>
> Concern #5: conditional invariant in P(X|Y). Learning an conditional invariant in P(X|Y) was first proposed in Zhang 2013, and later extended to deep neural networks in  Li 2018. Our work is based on this foundation and extended it to the continual learning setting.

---

### Official Review · AnonReviewer1 · 2020-10-29
**New problem setting but technical contribution is not significant**

**Rating:** 5
**Confidence:** 4

**Review:**

This paper investigates continual learning under domain shift and proposes a conditional invariant experience replay (CIER) method accordingly. CIER aims to retain old knowledge, acquire new information, and generalize to unseen domains. In particular, CIER uses adversarial training to correct the domain shift. Experimental results on three benchmark datasets are reported and discussed.

Pros.
1. The problem setting of continual learning with domain shift is well motivated. The authors explained the rationality of this new problem setting by using a causal model.
2. Three scenarios are considered and evaluated, including the significant shift, mild shift and no shift. Experimental results show that the proposed methods outperform baselines in many cases.
3. Overall the paper is well written and clearly organized. The technical details are easy to follow.

Cons.
1. Although the problem setting is new, my main concern is the limited novelty of the proposed CIER method. In detail, experience replay has been extensively studied in continual learning, while the domain adversarial learning has also been widely used in domain adaptation literature for many years. The proposed CIER method, as illustrated in Figure 2, simply combines these two components.
2. Ablation studies of the proposed method are missing. Thus, it is difficult to understand the contribution of each component, the effects of different sampling buffer sizes, etc.

---

> ### Author Response · Authors · 2020-11-24
> **Response to Reviewer 1**
>
> We thank the Reviewer for your comments. Please find our response below.
> Concern #1: limited novelty. Our main goal in this work is establishing new benchmarks for continual learning. We first show that existing state-of-the-art methods struggle to perform well on unseen domains. Then, we propose a simple strategy to learn an invariant representation to benefit experience replay in all settings with various degrees of domain shift.
>
> Concern #2: missing ablation studies. Our method only adds the invariant learning component to
> experience replay. Therefore, instead of conducting a separate ablation study section, we directly compare two versions of the domain discrepancy measurements (JSD and Wasserstein distances) in all experiments. We will consider other ablation studies regarding the hyper-parameters and features visualization in the future.

---

### Official Review · AnonReviewer4 · 2020-10-29
**Impractical set-up for continual learning**

**Rating:** 3
**Confidence:** 5

**Review:**

Summary: This paper talks about online continual learning in scenarios where there might be domain-shift during test time. Though I find the problem to be important, I believe that the solution proposed in this paper is straightforward (which is fine) and imposes a new set of constraints (knowing a priori the domain id during training) making the problem quite impractical for scenarios where continual learning might be useful. Please find below my comments in detail:

1. Problem formulation: I do agree with the authors that task-free set-up (also known as single-head) is something we should focus more on, and like the fact that they remove this constraint of knowing task id during train/test. However, while relaxing this constraint, they added a new assumption of knowing the domain id during training. I find this rather a more strict requirement. The right problem formulation would be where there is a stream of samples coming (online) with a relatively blurry task boundary, and these samples might belong to different domains. Knowing domains a priori is very impractical in continual learning set-up. Could you please scenarios where it’s feasible to know domain id a priori?

2. Approach: While I understand the ER part, I do not understand clearly section 4.1. What is the need of all GAN literature here? Why do you call D_j a domain critic? Am I wrong in saying that the only role of D_j for class y_j is to apply cross-entropy loss over domains (clipping if needed)? Please correct me if I am wrong. The section was a bit unclear to follow.

3. How do you make sure that the assumption of data balance is valid given that the memory is very small compared to the current dataset. Also, it seems like that the memory sizes of 9k, 10k, and 34k are too big. Please comment.

4. Training time: Since the set-up is online, training efficiently is extremely crucial. In Algorithm 1, every parameter update requires |Y| extra backprops for the domain critics SGD step. Could you please comment on the training time?

5. References: I would suggest the authors to correct their citations a bit. For example, when talking about task-free, I would also cite iCarl, synaptic intelligence (SI), and RWalk. Similarly, the forgetting measure used in this paper was not proposed in Chaudhry 2019a, it was proposed in RWalk.

Overall, even though the problem is important, I find the final experimental set-up to be impractical. There already are too many impractical formulations for continual learning, I would rather refrain myself from encouraging a new one. On top of this, there are a few technical aspects (training time, memory budget etc) I do not completely agree with, therefore, would request authors to answer above questions for clarity.

---

> ### Author Response · Authors · 2020-11-24
> **Response to Reviewer 4**
>
> We thank the Reviewer for your detailed feedback. Please find our response to your comments below.
> Concern #1: blurry task setting and example of knowing domain id a priori. We thank the Reviewer for suggesting the blurry task-boundary scenario, which is an interesting problem for future work.
> In this work, we only assume the domain label is given during training; at test time, only the observations (images) are given. This assumption is realistic since the domain identifier is just another supervision signal with the class label. A possible scenario for our formulation is learning a self-driving car agent where the first task is riding in urban areas and the second task is riding in rural areas. During training, we know the environment (urban/rural) and can provide this information to the agent. During testing, it is better to make predictions based only on the sensors' signals, without knowing the environment.
>
> Concern #2: GAN literature, the role of D_j. We address the domain shift by learning an invariant representation in P(X|Y), which leads to a minimax game similar to GAN loss.  When the model minimizes the Wasserstein distances between training domains, D_j does not predict the domains, but rather predicts a domain score; therefore, we followed the naming convention in WGAN to call D_j the critics. In the revision, we changed critic’s notation from D_j to K_j to improve the clarity as suggested by R3.
>
> Concern #3: data balance assumption, memory sizes are too big. Domain generalization methods do not make assumptions regarding the testing domains. Thus, the data balance assumption is needed to show that the methods will work.
> We apologize for the typos of the memory size. In all experiments, we consider a memory size equal to 100 samples per class, which translates to the total memory size of 900, 1000, and 34,500 in the iVehicle, iAnimal, and iDomainNet benchmarks. Using a memory of 100 samples per class is standard in continual learning. We fixed this typo in the revision.
>
> Concern #4: training time. We apologize for the confusion in our Alg.1. We want to clarify that the critics’ loss requires |Y| forward training passes but only one backward pass, as shown in Eq. 4,5. Moreover, this procedure only updates the critics’ parameters, which only have a small number of parameters. In the revision, we include the number of parameters and running time of our experiments Section 5.4.
>
> Concern #5: fixing references. Thank you for your suggestion, we corrected the reference accordingly in the revision.

---

### Official Review · AnonReviewer2 · 2020-11-07
**Official Blind Review #2**

**Rating:** 4
**Confidence:** 4

**Review:**

This paper studies the domain adaptation problem when the source data comes from multiple domains continuously and the test domain for adaptation is unknown. The assumption used for domain shift is that the domain label would change the features but not the labels. So the main idea is to learn invariant representations across all the domains and avoid spurious correlation on the domain label. The proposed method then involves a multiplayer minimax game. The adversaries are the domain discriminator for each class, which tries to maximize the domain discrepancy. The minimizer player is the representation learner. The paper introduces two discrepancy measure based on the Jensen-Shannon divergence and the one dimensional Wasserstein distance. In the experiment, the data set for continual learning is constructed using domain shift data such that it mimics the online learning setting. The results are competitive in comparison with a limited set of baselines.


Strong points:
1. The paper focused on an interesting and important topic.
2. The multiplayer adversarial game in terms of minimizing domain discrepancy seems to be novel.

Weak points:
1. The online or continual learning perspective is merely solved by keeping an episodic source data buffer, which I think is overly simplified. In general, I have a question about how this adversarial method would work when there is not an online/continual learning component. Given a fixed target, this a static set of source domains, it seems the method should be still valid. So I am not sure how the domain generalization side and the online side of the method interact with each other. Investigating the online setting before the batched setting seems problematic to me

2. The evaluation in experiments shows that the FM measure of the proposed method is not very competitive. I believe it is related to how the memory is sampled and the length of the task sequence. However, it also indicates the method is not very satisfactory for avoiding forgetting. Otherwise, it could also be the case the conditional domain shift assumption is not valid in the data.

3. The idea to learn an invariant representation is not novel. For example, the invariant risk minimization (IRM) method is exactly dealing with the same problem. Using an episodic source data buffer, it seems you can also apply IRM to solve the problem. I think it should be included as a baseline. In general, more invariant learning approaches should be discussed in the related work.

4. Writing can be significantly improved.

Given the weak points, I recommend rejection for this paper.

Here are some of my questions and additional suggestions:
1. I am curious about how the invariant feature looks like. Basically, after learning the adversarial domain predictors, how the representation learning looks like. It would be nice to have some visualization on that.

2. I do recommend focusing on a batched problem before going to the online version. I feel even though it is not the case that the batched method will work for online cases, at least in domain generalization we cannot expect a bad learner to work online when the data is even more limited. Even for an online paper, showing the performance for the batched version seems to be necessary.

3. How \alpha is chosen? And how it affects learning? In general, more ablation studies would improve paper quality.

---

> ### Author Response · Authors · 2020-11-24
> **Response to Reviewer 2**
>
> We thank the Reviewer for your detailed feedback and suggestions to improve the paper. Please find below our response.
>
> Concern #1: problem setting, prevent forgetting by an episodic memory. Thank you for your comment. Domain generalization in the batch setting (fixed target, static set of source domains) has been extensively studied in the literature as we discussed in Section 2.2. However, most works assume the targets (labels) and source domains are fixed from the beginning, which might have limited applications. On the other hand, continual learning methods address a more realistic problem when new labels can arrive overtime, but they do not consider the domain changes between training and testing domains. Thus, we aim at bridging the gap between these two problems in our study.
>
> Concern #2: FM measure of the method is not very competitive, the conditional domain shift assumption is not valid. Thank you for your comment. We want to remind that our experiments were in the online, task-free setting, which is the most challenging continual learning protocol. Therefore, it is unsurprising that the FM of all methods, including state-of-the-art such as ER and MIR, are high.  Moreover, we carefully constructed the benchmarks so that the domain shift assumptions hold: training and testing domains are different and there is no class-imbalance between training and testing data, we provided some illustrative examples in Appendix C.
> The results in Table 2 and 3 show that domain shift does not affect the forgetting but directly affects the model’s ability to perform the tasks: FM values are consistent across three degrees of domain shift, only ACC and LA values are significantly different.
>
> Concern #3: learning an invariant representation is not novel, baseline IRM.  Thank you for your recommendation. We included more representation learning methods and will consider more baselines such as IRM in our revision.
>
> Concern #4: writing can be improved. Thank you for your suggestion, we significantly improved the writing in the revision.
>
> Concern #5 invariant features. Thank you for your suggestion. However, all models’ performances are quite low and hence, the features are not well separated enough to be clearly visualized in a plot. Therefore, we will leave this for future work.
>
> Concern #6: batch problem. Thank you for your comment. Since we want to demonstrate a novel problem setting of domain shift in continual learning, we consider the online setting (tasks arrive online and data within each task are also online) to show that many CL methods cannot perform well on unseen domains. However, our benchmarks can be easily benchmarked in the batch CL settings in the future.
>
> Concern #7: \alpha values. In this work, we follow the common practice to use a small amount of data of each task as the validation set to cross-validate the hyper-parameters. We cross-validated $\alpha$ from 0.95 to 0.05 and observed that  $\alpha =0.95$ yielded the best results.

---

### Decision · Program_Chairs · 2021-01-07
**Final Decision**

**Decision:**

Reject

**Comment:**

This paper presented an online continual learning method where there may be a shift in data distribution at test time. The paper proposes a Conditional Invariant Experience Replay (CIER) approach to correct the short which matches the distribution of inputs conditioned on the outputs. This is based on an adversarial training scheme.

The reviewers found the problem setting interesting but found the approach to be lacking in novelty and problem formulation somewhat restrictive (e.g.,  requiring domain id during training). The author feedback was taken into account but the reviewers stayed with their original assessment and, even after the rebuttal phase, none of the reviewers is in favor of accepting the paper.

The authors are advised to consider the feedback from the reviewers which will hopefully help to improve the paper for a future submission to another venue.